# Early Immune Modulation in European Seabass (*Dicentrarchus labrax*) Juveniles in Response to *Betanodavirus* Infection

Mariana Vaz [1], Damiana Pires [1], Pedro Pires [1], Marco Simões [1], Ana Pombo [1,2], Paulo Santos [3], Beatriz do Carmo [1], Ricardo Passos [1], Janina Z. Costa [4], Kim D. Thompson [4] and Teresa Baptista [1,2,*]

[1] MARE—Marine and Environmental Sciences Centre, Polytechnic of Leiria, Edifício CETEMARES, Av. Porto de Pesca, 2520-620 Peniche, Portugal; mariana.c.vaz@ipleiria.pt (M.V.); damiana.pires@ipleiria.pt (D.P.); pedro.pires@ipleiria.pt (P.P.); marco.a.simoes@ipleiria.pt (M.S.); ana.pombo@ipleiria.pt (A.P.); beatriz.carmo@ipleiria.pt (B.d.C.); ricardo.passos@ipleiria.pt (R.P.)

[2] School of Tourism and Maritime Technology, Polytechnic of Leiria, Campus 4—Rua do Conhecimento no 4, 2520-641 Peniche, Portugal

[3] Centro Interdisciplinar de Investigação Marinha e Ambiental (CIIMAR), University of Porto, Terminal de Cruzeiros do Porto de Leixões, Av. General Norton de Matos S/N, 4450-208 Porto, Portugal; paulo.santos@ciimar.up.pt

[4] Moredun Research Institute, Pentlands Science Park, Bush Loan, Penicuik EH26 0PZ, UK; janina.costa@moredun.ac.uk (J.Z.C.); kim.thompson@moredun.ac.uk (K.D.T.)

[*] Correspondence: teresa.baptista@ipleiria.pt; Tel.: +351-919-410-798

**Abstract:** The early host–pathogen interaction between European seabass (*Dicentrarchus labrax*) and *Betanodavirus* was examined by using juvenile fish infected intramuscularly with RGNNV (red-spotted grouper nervous necrosis virus). The time course selected for sampling (0–144 h post-infection (hpi)) covered the early stages of infection, with hematological, antioxidant and immunological responses examined. Early activation of the host's immune system was seen in the first few hours post-infection (6 to 9 hpi), as evidenced by an increase in *tnfα*, *cd28* and *c3* expression in the head kidney of infected fish. Most hematological parameters that were examined showed significant differences between sampling times, including differences in the number of thrombocytes and various leukocyte populations. The plasma lysozyme concentration decreased significantly over the course of the trial, and most antioxidant parameters examined in the liver showed significant differences over the infection period. At 144 hpi, peak expression of *tnfα* and *il-1β* coincided with the appearance of disease symptoms, peak levels of virus in the brain and high levels of fish mortality. The results of the study show the importance of analyzing the early interactions between European seabass and *Betanodavirus* to establish early indicators of infection to prevent more severe outcomes of the infection from occurring.

**Keywords:** European seabass; nervous necrosis virus; *Betanodavirus*; hematology; oxidative stress; fish immunity



## 1. Introduction

Intensification of marine fish production has resulted in high stocking densities and stressful culture conditions for fish, thus contributing to increased disease outbreaks within aquaculture farming systems. Viral diseases are an important obstacle for aquaculture development, due to the high economic impact associated with disease outbreaks [1]. One viral disease that has been a major problem for the aquaculture industry is viral encephalopathy and retinopathy (VER), also known as viral nervous necrosis (VNN), caused by nervous necrosis virus (NNV) [2]. NNV belongs to the family Nodaviridae, genus *Betanodavirus* [3]. It is a non-enveloped virus, with a diameter of 20–25 nm [4], constituting two single-stranded positive-sense RNA segments, with RNA1 encoding RNA-dependent RNA polymerase, and RNA2 encoding the capsid protein [4,5]. Of the four

NNV genotypes affecting fish, RGNNV (red-spotted grouper nervous necrosis virus) is widely distributed in Southern Europe, across the Asian coast and also in Australia [2,6,7]. The clinical signs observed depend on the fish species, the age and the developmental stage of the fish [2,8], but they usually include spiral swimming, looping behavior, exophthalmia and eye opacity [7]. *Betanodavirus* outbreaks mainly affect larval and juvenile stages, with mortality levels as high as 100% [7,9] in European seabass (*Dicentrarchus labrax*) [10], seabream (*Sparus aurata*) [11] and Atlantic cod (*Gadus morhua*) [1,12].

Understanding virus pathogenicity is important for the development of therapeutic and prophylactic measures to mitigate against pathogen proliferation [13]. The first line of defense against the virus is produced by the host's innate immune system [14], which produces a rapid and non-specific response, which is mediated by type I and II interferons (IFN), as reported to occur in NNV-infected European seabass [15]. The adaptive immune system, on the other hand, responds specifically to the pathogen, producing a highly specific and long-lasting immune response [16,17].

Early detection of the viral infection in fish is an important strategy to identify potential disease outbreaks before they occur, and it may be possible to identify biomarkers of early infection by assessing the host's response to the developing viral infection. Although several studies have reported on the dynamics of NNV infection and associated virulence factors of the virus [7,13,15], few studies have examined early activation of the host's response to the virus immediately after infection. The aim of this study was to evaluate the hematological, antioxidant and immunological and molecular responses in European seabass juveniles shortly after NNV infection in order to understand these responses and identify possible biomarkers that could help in an early diagnosis of the disease.

## 2. Materials and Methods

### 2.1. Time-Course Infection with RGNNV

The study was conducted in accordance with "The guidelines on animals protection used for scientific purposes from the European Directive 2010/63/EU" under project permission 0421/000/000/2019. A time-course study was performed at CETEMARES facilities (Polytechnic of Leiria, Peniche, Portugal), using 156 European seabass juveniles ($31.2 \pm 7.2$ g). Fish were acclimated in a 2000 L tank at 25 °C and sampled to confirm that they were NNV-free. After two weeks, fish were randomly distributed into nine recirculating seawater systems, i.e., three systems per experimental condition (21 fish per tank), and three (10 fish per tank) for monitoring mortality. The animals received an intramuscular (IM) injection of either 100 μL phosphate buffered saline (PBS) pH 7.3 as the control group or 100 μL of RGNNV at $10^7$ TCID$_{50}$ mL$^{-1}$ (strain ERV378/03) according to Reference [18]. The fish were maintained at 25 °C, with aeration, and were fed *ad libitum* with a commercial diet. Mortality, temperature, salinity and dissolved oxygen were monitored daily. Two animals per tank ($n$ = 6 per treatment) were sampled at 0 (pre-challenge), 6, 9, 24, 48, 120 and 144 h post-infection (hpi). Fish were anesthetized with 2-phenoxyethanol (VWR Chemicals, Radnor, PA, USA) (0.5 mL L$^{-1}$), blood sampled and euthanized with an overdose of anesthetic before liver, head-kidney and brain-tissue collection. Blood was sampled from the caudal vein with heparinized syringes and stored in microtubes containing 20 μL of heparin (3000 U). After hematological parameters analysis, the remaining blood was centrifuged ($10,000 \times g$ at 4 °C for 10 min) for plasma collection and was stored at −80 °C until the analysis of innate humoral parameters (lysozyme concentration, peroxidase activity, antiproteases, proteases activity and production of nitric oxide). The liver, collected for determination of oxidative stress parameters (lipid peroxidation, catalase, superoxide dismutase, total glutathione and glutathione-S-transferase), was snap-frozen in liquid nitrogen after collection and stored at −80 °C until analyzed. Head kidney and brain (collected at 0 h pre-challenge, 48 and 144 hpi) were preserved in RNAlater (Sigma Aldrich, St. Louis, MO, USA) for 24 h at 4 °C; the RNAlater was decanted, and the samples stored at −80 °C until analysis for gene expression or virus quantification, respectively.

## 2.2. Hematological Parameters

The hematological profile of the European seabass was carried out by determining total red blood cell (RBC) and white blood cell (WBC) counts, using a Neubauer chamber, hemoglobin concentration (hemoglobin, SPINREACT, Spain) and hematocrit values (Ht). Differential WBC counts were made from blood smears that had been air-dried and fixed with formaldehyde–ethanol solution (90% absolute ethanol with 3.7% formaldehyde) for 1 min [19]. Neutrophils were identified according to Reference [20], and 200 leucocytes were counted, from which the percentages of neutrophils, monocytes, lymphocytes and thrombocytes were determined. Blood indexes, including mean corpuscular volume (MCV), mean corpuscular hemoglobin (MCH) and mean corpuscular hemoglobin concentration (MCHC), were determined according to Reference [21].

## 2.3. Innate Immune Humoral and Antioxidant Parameters

Plasma lysozyme concentrations were determined by using a turbidimetric assay, as described by Costas et al. [22], and calculated from a standard curve. Plasma peroxidase activity was determined according to Quade and Roth [23], with one unit of peroxidase representing an absorbance change of 1.0 OD at 450 nm.

The activity of antiproteases was determined according to Machado et al. [21], by measuring trypsin inhibition in plasma. Thus 10 µL of plasma and 10 µL of trypsin in $NaCO_3$ (5 mg mL$^{-1}$) were added and incubated for 10 min at room temperature (RT). Then, 100 µL of PBS (13.9 mg mL$^{-1}$, pH 7.0) and 125 µL of azocasein (20 mg mL$^{-1}$ in $NaHCO_3$, pH 8.3) were added and incubated for 1 h, at RT. The reaction was stopped with the addition of 250 µL of trichloroacetic acid (TCA) (100 mg mL$^{-1}$) and subsequently incubated for 30 min at RT. The mixture was then centrifuged ($10,000 \times g$ for 5 min), and the supernatant was collected and added to 100 µL of NaOH (40 mg mL$^{-1}$). Finally, samples were read in duplicate at 450 nm. The activity of proteases was also quantified in plasma by measuring the level of uninhibited trypsin according to Reference [21]. For this analysis, 10 µL of plasma, 100 µL of PBS (13.9 mg mL$^{-1}$, pH 7.0) and 125 µL of azocasein (20 mg mL$^{-1}$ in $NaHCO_3$, pH 8.3) were added and incubated for 24 h, at RT. The reaction was stopped with the addition of 250 µL of TCA (100 mg mL$^{-1}$) and by following the same procedure as for the activity of anti-proteases, with the samples read at 450 nm.

The production of nitric oxide in plasma was measured according to Taffala et al. [24]. A total of 25 µL of plasma, 100 µL of 1% sulfanilamide solution in 2.5% phosphoric acid and 100 µL of *N*-naphthylethylenediamine solution in 2.5% phosphoric acid were mixed, and after a 10-min incubation at RT, the samples were read at 540 nm. The concentration of nitrite (µMol mL$^{-1}$) in the samples was determined by using a standard curve with known concentrations of sodium nitrite.

Oxidative stress biomarkers were measured in the liver samples. Liver was homogenized with 10 volumes of ultrapure water, and then a 200 µL aliquot of the homogenate was placed in a microtube with 4 µL of 4% BHT (2,6-Di-tert-butyl-4-methylphenol) in methanol. The tissue homogenate was mixed 1:1 (*v*/*v*) with potassium phosphate buffer (0.2 M, pH 7.4) and centrifuged at $10,000 \times g$ at 4 °C for 20 min. The supernatant was stored in a new microtube at −80 °C until analysis of the oxidative-stress biomarkers.

The total protein concentration of the liver homogenate was determined by using a commercial BCA Protein Assay Kit (Pierce™, Waltham, MA USA). Liver catalase (CAT) activity was measured according to the method described by Clairborne [25], adapted as a microplate assay [26], while liver lipid peroxidation (LPO) was determined according to Bird and Draper [27]. Liver glutathione-s-transferase (GST) activity was determined as reported by Habig et al. [28] and adapted as a microplate assay by [29]. Superoxide dismutase (SOD) was determined as described by Almeida et al. [30]. Total glutathione (tGSH) was measured according to Baker et al. [31] and Rodrigues et al. [26]. Total albumin concentration in the plasma was analyzed with a BCG Albumin Assay Kit (Sigma-Aldrich, St. Louis, MO, USA), according to the manufacturer instructions.

Unless otherwise stated, all chemicals were purchased from Sigma-Aldrich (St. Louis, MO, USA).

### 2.4. Viral Quantification

Viral RNA from virus supernatant (with a known $TCID_{50}$ $mL^{-1}$) and brain were extracted by using the NucleoSpin RNA Mini kit (Macherey-Nagel, Düren, Germany), according to the manufacturer's instructions. The yield of RNA obtained was determined by using a NanoDrop 2000 (ThermoFisher Scientific, Waltham, MA, USA), and its quality and integrity were verified by electrophoresis (Wide Mini-Sub Cell GT, Bio-Rad, California, CA, USA) on 2% agarose gel.

NNV quantification was performed by using a SuperScript III Platinum One-Step qRT-PCR kit (Invitrogen, Carlsbad, CA, USA) with a final volume of 10 µL. Primers used were Noda Taq1-FW (5′-CAACTGACARCGAHCACAC–3′) and Noda Taq1-RV (5′-CCCACCAYTTGGCVAC–3′) and a specific probe for *Betanodavirus* (5′-6FAM-CARGCRAC-TCGTGGTGCVG-BHQ1-3′) [10], at a final concentration of 10 µM. The samples and standard curve were prepared in triplicate. The standard curve was prepared with ten-fold serial dilutions [18]. The CFX Connect™ Real-Time System (Bio-Rad, California, USA) was used with the following conditions: reverse transcriptase—10 min at 55 °C, 5 min at 95 °C; and 40 cycles of 15 s at 95 °C and 1 min at 60 °C. With the threshold cycle (Ct) obtained, the amount of virus in the fish brains was calculated by using the equation of the line [x = (yc)/m] determined with the standard curve Ct values plotted against $TCID_{50}$.

### 2.5. Immune Gene Expression

RNA was extracted from the head kidney by using an NZY Total RNA isolation kit (NZYTech, Lisbon, Portugal), according to the supplier's instructions. Extracted RNA was evaluated by spectrophotometry (NanoDrop 2000, ThermoFisher Scientific, Waltham, MA, USA), and the quality and integrity of RNA were verified as described above. Then cDNA synthesis was performed on a T100™ Thermal Cycler (Bio-Rad, California, CA, USA), following the instructions of the NZY First-Strand cDNA Synthesis kit (NZYTech, Lisbon, Portugal).

The immune genes selected for analysis included endogenous Mx protein (MXA), interferon regulatory factor 3 (IRF3), complement component (C3), T-cell-specific surface glycoprotein (CD28), tumor necrosis factor (TNFα) and interleukin 1β (IL-1β). The housekeeping gene, β-actin (ACTB), was used for gene normalization. Published primers for the selected immune genes are shown in Supplementary Table S1. The efficiency of the primers was evaluated through linear regression, taking into account the slope and the average threshold cycle (Ct) obtained. CFX Connect™ Real-Time System (Bio-Rad, California, CA, USA) was used for the immune-gene quantitative PCR (qPCR). The qPCR reaction was performed with 1 µL of cDNA (5-fold dilution factor), 0.3 µL of each primer (reverse and forward, at 10 µM each) and 5 µL of iTaq™ Universal SYBR® Green Supermix (Bio-Rad, California, CA, USA) at a final volume of 10 µL. Standard cycle conditions were 95 °C for initial denaturation for 3 min, followed by 40 cycles of 95 °C for 30 s for denaturation and annealing/extension for 20 s with temperature-specific conditions for each set of primers (Supplementary Table S1). All the samples were analyzed in triplicate. The samples were normalized by using the housekeeping gene, and gene expression was determined according to Pfaffl [32].

### 2.6. Statistical Analysis

Hematological data, innate immune and antioxidant parameters, and gene expression are presented as the mean of six replicates ± standard deviation (SD). Data from each parameter were tested for significant statistical differences, using a one-way analysis of variance (ANOVA), with sampling time as a factor, followed by multiple comparison's tests—Tukey's test, whenever assumptions of homogeneity and normality were met. For virus quantification, a *t*-test was performed. In cases where data did not meet homogeneity

and normality assumptions, a Kruskal–Wallis test was performed. Statistical significance was tested at $p < 0.05$, and all statistical analysis was performed by using IBM's SPSS software (version 27, USA).

## 3. Results

### 3.1. Mortality

The level of cumulative mortality obtained in the experimentally infected European seabass is presented in Figure 1. No mortality was observed in the control group. Typical clinical signs of VER, exophthalmia and fast looping were observed in the infected group with 33.3% mortality 5 days post-infection (dpi), and reaching maximum mortality of 86.7% by 9 dpi.

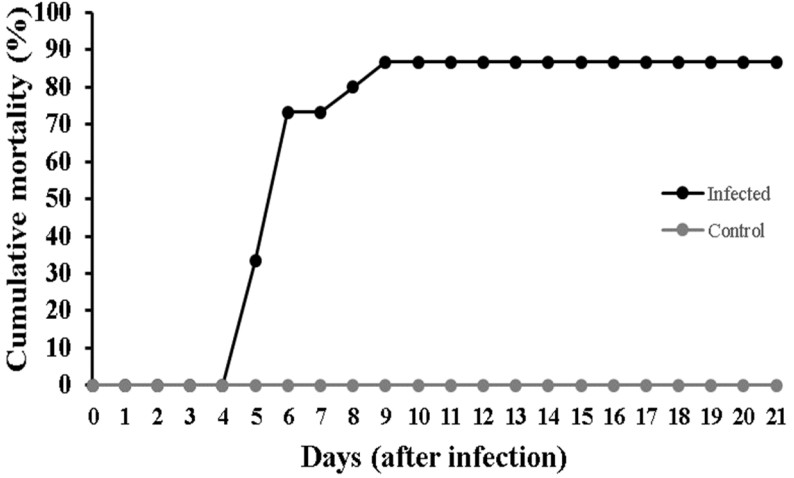

**Figure 1.** Percentage of cumulative mortality recorded in European seabass (*Dicentrarchus labrax*) control and RGNNV-infected fish, using triplicate tanks of ten fish per group.

### 3.2. Hematological Parameters

The hematological profile of European seabass infected with RGNNV is presented in Table 1. Most parameters showed significant differences between sampling times ($p < 0.05$), except for the total erythrocyte counts, hematocrit % and MCV. Leucocyte and thrombocytes counts decreased significantly at 9 hpi. Neutrophil numbers increased at almost every sampling point, except at 48 hpi, and monocytes displayed a similar trend, but they did not increase at 9 hpi. Lymphocytes decreased at 6 hpi and returned to basal values at 48 hpi, only to decrease again at 144 hpi. Hemoglobin concentration increased at 24 hpi, accompanied by the increase in MCH and MCHC values at the same sampling time point.

**Table 1.** Hematological parameters of European seabass (*Dicentrarchus labrax*) infected with RGNNV at different sampling times post-infection.

| | Time (hpi) | | | | | | |
|---|---|---|---|---|---|---|---|
| | 0 | 6 | 9 | 24 | 48 | 120 | 144 |
| RBC ($10^6 \ \mu L^{-1}$) | $2.49 \pm 0.53$ | $2.70 \pm 0.48$ | $3.20 \pm 0.12$ | $2.53 \pm 0.56$ | $2.96 \pm 1.02$ | $2.63 \pm 0.49$ | $2.27 \pm 0.31$ |
| WBC ($10^4 \ \mu L^{-1}$) | $6.17 \pm 1.21$ [a] | $5.02 \pm 1.19$ [ab] | $3.92 \pm 1.99$ [b] | $4.98 \pm 0.35$ [ab] | $4.57 \pm 1.40$ [ab] | $5.37 \pm 0.94$ [ab] | $6.93 \pm 2.00$ [a] |
| Neutrophils ($10^4 \ \mu L^{-1}$) | $0.09 \pm 0.08$ [a] | $1.04 \pm 0.43$ [bc] | $0.35 \pm 0.19$ [b] | $0.35 \pm 0.05$ [b] | $0.18 \pm 0.12$ [a] | $0.58 \pm 0.22$ [bc] | $1.40 \pm 0.44$ [c] |
| Monocytes ($10^4 \ \mu L^{-1}$) | $0.07 \pm 0.06$ [a] | $0.28 \pm 0.13$ [b] | $0.09 \pm 0.09$ [ac] | $0.19 \pm 0.07$ [bc] | $0.07 \pm 0.05$ [ac] | $0.21 \pm 0.11$ [bc] | $0.94 \pm 0.57$ [b] |

**Table 1.** *Cont.*

| | | | | Time (hpi) | | | |
|---|---|---|---|---|---|---|---|
| Lymphocytes ($10^4$ μL$^{-1}$) | 2.29 ± 0.47 [a] | 1.11 ± 0.26 [b] | 1.22 ± 0.58 [b] | 1.34 ± 0.37 [b] | 1.76 ± 0.36 [ab c] | 2.25 ± 0.24 [ac] | 1.48 ± 0.83 [bc] |
| Thrombocytes ($10^4$ μL$^{-1}$) | 3.81 ± 0.85 [a] | 2.60 ± 0.79 [ab] | 2.05 ± 1.01 [b] | 3.29 ± 0.58 [ab] | 2.55 ± 1.06 [ab] | 2.56 ± 0.42 [ab] | 3.12 ± 0.99 [ab] |
| Hb (g dL$^{-1}$) | 1.85 ± 0.70 [a] | 2.57 ± 0.66 [ab] | 2.37 ± 0.50 [ab] | 3.36 ± 1.21 [b] | 2.79 ± 0.93 [ab] | 2.66 ± 0.34 [ab] | 2.92 ± 0.50 [ab] |
| Ht (%) | 25.45 ± 3.59 | 25.80 ± 1.48 | 22.80 ± 2.39 | 25.20 ± 3.70 | 25.50 ± 3.11 | 26.33 ± 3.39 | 22.50 ± 3.67 |
| MCV (μm$^3$) | 89.81 ± 9.60 | 92.52 ± 2.93 | 75.19 ± 1.60 | 103.68 ± 21.3 | 114.10 ± 83.5 | 95.06 ± 10.56 | 91.97 ± 17.79 |
| MCH (pg cell$^{-1}$) | 7.27 ± 2.31 [a] | 9.60 ± 2.25 [ab c] | 7.33 ± 1.70 [ac] | 13.70 ± 4.88 [b] | 11.55 ± 9.3 [ab c] | 10.30 ± 1.58 [b] | 12.01 ± 2.72 [b] |
| MCHC (g 100 mL$^{-1}$) | 8.04 ± 0.53 [a] | 10.22 ± 1.67 [ab] | 10.25 ± 1.13 [ab] | 13.52 ± 4.47 [b] | 12.57 ± 0.74 [b] | 10.14 ± 1.16 [ab] | 13.04 ± 1.61 [b] |

Values in the same row with different letters show statistical differences ($p < 0.05$). Values are represented as means ± standard deviation ($n = 6$). One-way ANOVA was performed to compare between sampling times. RBC, red blood cells; WBC, white blood cells; Hb, hemoglobin; Ht, hematocrit; MCV, mean corpuscular volume; MCH, mean corpuscular hemoglobin; MCHC, mean corpuscular hemoglobin concentration; hpi, hours post-infection.

### 3.3. Innate Humoral Parameters and Biomarkers

The immune parameters evaluated in European seabass infected with RGNNV are presented in Figure 2. The plasma lysozyme concentration decreased significantly over the course of the trial, from pre-challenge to 144 hpi ($p < 0.05$), while plasma peroxidase, antiproteases and proteases activity did not show any significant differences related to the infection ($p > 0.05$). Nitric oxide, however, showed a significant increase at 24 and 144 hpi ($p < 0.05$).

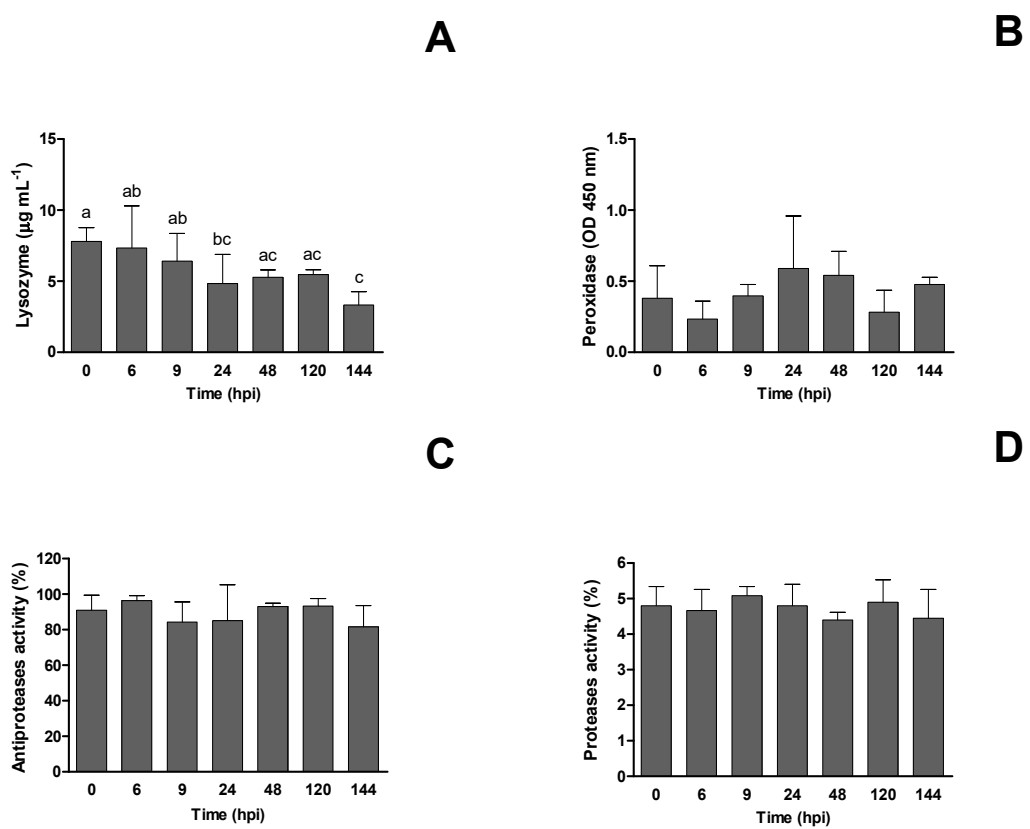

**Figure 2.** *Cont.*

**E**

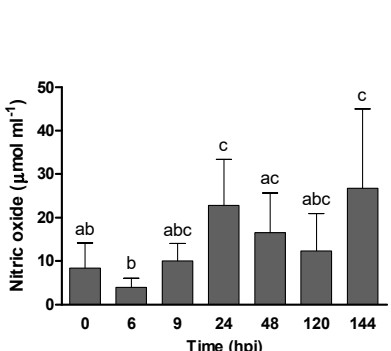

**Figure 2.** Innate humoral parameters of European seabass (*Dicentrarchus labrax*) infected with RGNNV at different hours post-infection (hpi): (**A**) lysozyme concentration, (**B**) peroxidase activity, (**C**) antiproteases activity, (**D**) proteases activity and (**E**) nitric oxide are presented as mean ± SD (*n* = 6). One-way ANOVA was used to compare between different sampling times. Different letters show statistically significant differences between sampling times ($p < 0.05$).

The antioxidant activity in the liver and albumin in the blood of European seabass infected with RGNNV are presented in Figure 3. All parameters showed significant differences between sampling times ($p < 0.05$), except for SOD activity and plasma albumin concentration ($p > 0.05$). The CAT started to decrease after 9 hpi and remained lower until the final sampling time at 144 hpi. The GST activity was reduced at 144 hpi when compared to values at 9, 24 and 48 hpi. A similar trend was observed for tGSH, but values were also reduced at 120 h when compared to 9, 24, and 48 hpi. The LPO increased after 48 hpi until the end of the sampling period.

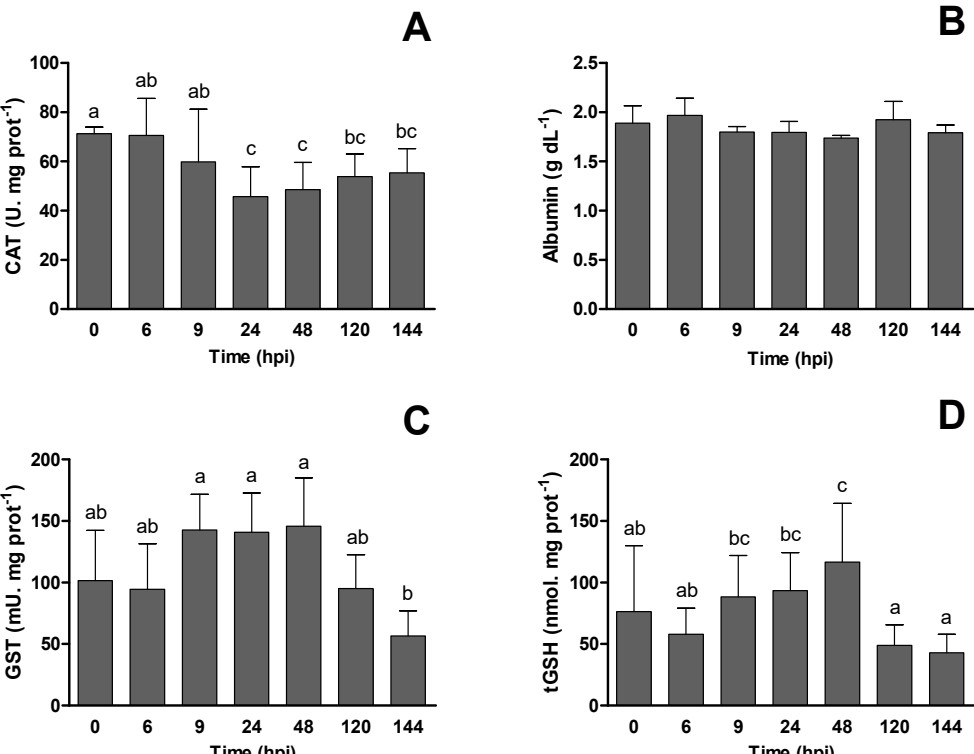

**Figure 3.** *Cont.*

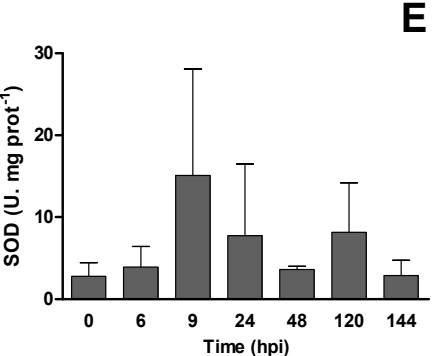
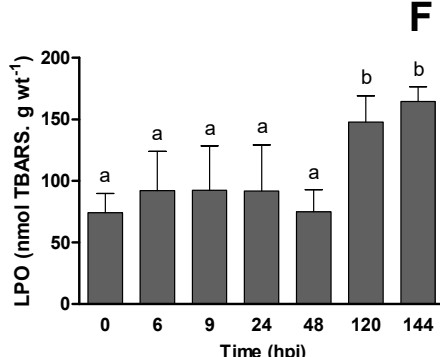

**Figure 3.** Antioxidant activity in the liver and albumin plasma concentration in European seabass (*Dicentrarchus labrax*) infected with RGNNV at different sampling times post-infection (hpi): (**A**) catalase activity (CAT), (**B**) albumin, (**C**) glutathione-s-transferase activity (GST), (**D**) total glutathione (tGSH), (**E**) superoxide dismutase (SOD) and (**F**) lipid peroxidation (LPO). Values are presented as mean $\pm$ SD (*n* = 6). One-way ANOVA was used to compare between different sampling times. Different letters show statistically significant differences between sampling times ($p < 0.05$).

### 3.4. Viral Quantification

A standard curve was prepared by using serial dilutions of viral RNA to establish a linear regression between mean Ct values and dilutions, with a starting concentration of NNV at $TCID_{50} = 10^7$ mL$^{-1}$ (Supplementary Figure S1). The quantification of the virus in unknown samples was determined from the following equation: y = $-3.5422$x + 36.407, with $R^2 = 0.9945$.

The quantification of NNV in brain tissue was determined at 48 and 144 hpi. Virus was not detected at 0 hpi prior to infection. There was a significant increase ($p < 0.05$) in the amount of virus detected at 48 ($8.24 \times 10^5$ $TCID_{50}$ mL$^{-1}$) and 144 hpi ($1.18 \times 10^7$ $TCID_{50}$ mL$^{-1}$). No virus was detected in the control fish.

### 3.5. Immune Genes Expression

The expression of immune genes in the head kidney of European seabass was evaluated after infecting the fish with NNV. At 48 hpi, a significant increase in *mxA* gene expression was observed, and then it decreased until the end of the sampling period at 144 hpi (Figure 4A). *irf3* expression peaked at 6 hpi and then decreased from that point until 24 hpi, with the expression between 6 and 24 hpi being significantly lower. Maximum expression was observed at 48 hpi (Figure 4B). Expression in *c3* decreased during the first 9 hpi from its basal levels (0 hpi) to almost being undetectable, whereas it increased significantly thereafter, remaining almost constant until the last sampling point, when maximum expression was seen (Figure 4C). *cd28* expression significantly increased up to 9 hpi, returned to baseline at 48 hpi, and then remained constant until the end of sampling (Figure 4D). *tnfα* expression significantly decreased at 48 hpi when compared to baseline values, and then it increased again significantly at 144 hpi (Figure 4E). Regulation of the *il-1β* gene increases significantly at 6 hpi, stabilizing its gene expression until 24 hpi, then decreasing at 48 hpi, and increasing again at 144 hpi, when the maximum expression of *il-1β* was observed (Figure 4F).

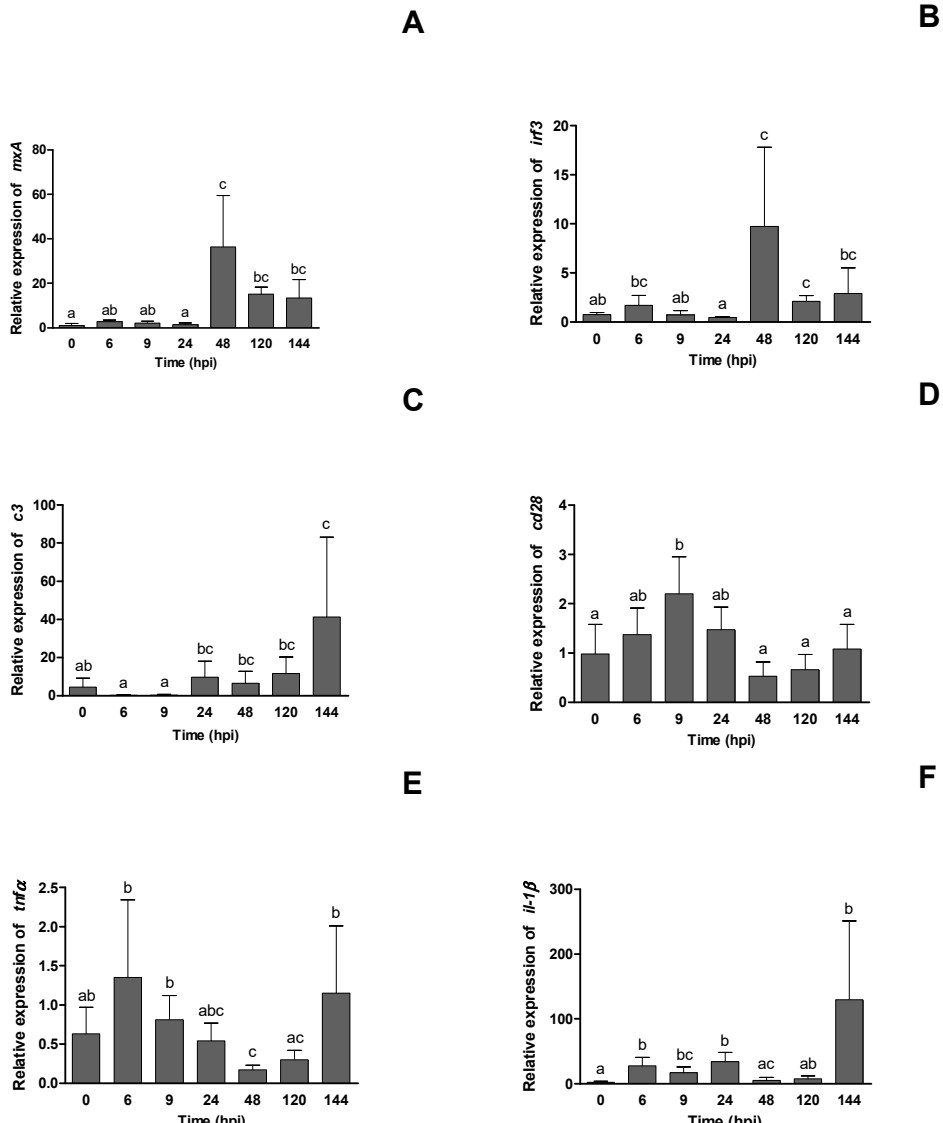

**Figure 4.** Relative expression of selected immune genes: (**A**) endogenous Mx protein, *mxA*; (**B**) interferon regulatory factor 3, *irf3*; (**C**) complement component, *c3*; (**D**) T-cell-specific surface glycoprotein, *cd28*; (**E**) tumor necrosis factor alpha, *tnfα*; and (**F**) interleukin 1 beta, *il-1β*, in the head-kidney samples at 0 h (pre-challenge), 6, 9, 24, 48, 120 and 144 h post-infection (hpi) with NNV of European seabass. Values are presented as mean $\pm$ SD (*n* = 6). One-way ANOVA was executed to compare between different sampling times. Different letters show statistically significant differences between sampling times ($p < 0.05$).

## 4. Discussion

We examined the early host–pathogen interaction between European seabass juveniles and RGNNV in order to understand when distinct host defense mechanisms are activated against the virus. Infection-related mortalities started from 5 dpi, with peak mortalities occurring by 9 dpi. The time for the first mortalities to be observed was in line with other studies [33,34]. The duration of experimental infection was extended beyond the sampling period; however, it was based on the challenge model we used [35]. In accordance with the literature [36], seven band grouper (*Hyporthodus sptemfasciatus*) infected IM with $10^{2.5}$ TCID$_{50}$ 100 μL$^{-1}$ fish$^{-1}$ recorded a peak mortality of 6 dpi (144 hpi). In this study, mortality started at 5 dpi, and at sampling period (6 dpi–144 hpi), the number of viral particles present in the brain ($1.18 \times 10^7$ TCID$_{50}$ mL$^{-1}$) was statistically higher when

compared to 48 hpi ($8.24 \times 10^5$ TCID$_{50}$ mL$^{-1}$), indicating a significant increase in the viral load and consequent infection in the host.

At 9 hpi, leukocytes and thrombocyte numbers were at their lowest level. This early decrease might indicate migration of the cells into the affected tissues at the infected site [37]. Both neutrophil and monocyte numbers displayed a similar trend, with numbers of cells increasing in the blood from 6 hpi and for most of the time thereafter. This rapid response by these cells is expected as part of the development of the inflammation process, since neutrophils are the first cells to move to a site of infection, followed by monocytes [38], explaining the higher number of these cells circulating as they move to the inflamed tissues. In the acute stage of an infection, a decrease in the number of lymphocytes is often seen, resulting in lymphopenia and being a sign of immune exhaustion [39]. Such a decrease in the lymphocytes population was observed on day 5 (120 hpi), when the first mortalities were observed, and a further drop was seen on day 6 (144 hpi), when mortality levels reached 73.3%. Hemoglobin and the associated indexes (MCH and MCHC) were augmented at 24 hpi, indicating that, at this time, oxygen was more efficiently transported in the blood, and with the infection undergoing, the capacity of the red blood cells to transport oxygen decreased.

Lysozyme, an innate humoral component, plays an important role in the primary line of defense against pathogens [40]. As the NNV infection progressed, lysozyme concentrations gradually reduced from 24 hpi onwards. This result is in line with immune depression previously noted by Reference [41], who described a similar relationship between lysozyme and lymphocytes levels being suppressed simultaneously. The fact that peroxidase activity was unchanged remains to be clarified, as it does not reflect the changes in neutrophil numbers observed. Nitric oxide has an active role in inhibiting virus replication, helping viral clearance and host recovery [42,43]. Its production can occur in neutrophils and monocytes [21,44], and it is critical for fish survival [45]. The production of nitric oxide in cells [45] may function as a response to pro-inflammatory cytokines, such as tumor necrosis factor $\alpha$ (TNF$\alpha$), bacterial lipopolysaccharide (LPS), parasites [46] and viral infections [47]. Infection of *Scophthalmus maximus* renal macrophages with viral hemorrhagic septicemia virus (VHSV) in vitro [46] resulted in the production of nitric oxide induced by the viral infection. As seen in the present study, there was a significant increase in nitric oxide production at 24 and 144 hpi, suggesting that this increase may be involved in inhibiting viral replication [48]. These results indicate that, in the first hours after viral infection, nitric oxide is produced by neutrophils and monocytes, with peak production at 144 hpi.

Antioxidant enzyme activity is a defense mechanism against oxidative stress and is considered to be part of the innate immune response. In the case of CAT, this activity was reduced from 9 hpi onwards, showing a decreased capacity to deal with oxidative imbalance, which can have a detrimental effect on other cellular processes [49]. GST levels increased and then decreased over the course of the infection, with the lowest levels observed at 144 hpi, reflecting the activity of tGSH, which disrupts glutathione-based antioxidant systems and further increases oxidative stress [50]. The inhibition of both enzymatic and non-enzymatic antioxidant systems is reflected by lipid peroxidation (LPO). Lipid peroxidation was significantly higher at 120 and 144 hpi, indicating the time when cells started to become damaged by an oxidative imbalance, and which corresponded with the mortality starting at 120–144 hpi.

After infection with *Betanodavirus*, viral particles enter the host cell's cytoplasm and initiate transcription and translation through the release of their viral genome [1]. This results in a "signaling cascade" of intercellular molecules [17] that activates an antiviral response by the host. Endogenous Mx protein (Mx) interacts with RNA-dependent RNA polymerase (RdRp) right after its translation in an attempt to degrade the viral protein through the process of autophagy [51,52]. Mx plays an important role in the control of NNV infection, as reported in grouper (*Epinephelus coioides*) [53]. In the present study, a significant increase in *mxA* gene expression was noted after 48 hpi, suggesting that the host's defenses had detected the presence of the virus and were trying to inhibit viral

replication through the production of Mx. The expression of *mxA* was previously observed from 24 hpi in the brain and head kidney of European seabass and Gilthead seabream, using the same viral dose as we used in our study ($10^6$ TCID$_{50}$ of NNV) [54]; however, *mxA* expression was still at basal levels at this time post-infection in our study. We also saw further upregulation of this gene at 120 and 144 hpi, in accordance with other study [14] whose authors observed a significant upregulation of *mx* expression in the head kidney of European seabass infected with NNV up to 168 hpi. On the other hand, the onset of *mx* expression in fish infection with RGNNV [55,56] appears to be related to their life stage, the viral isolates and method of infection, ranging from 6 to 72 hpi.

Interleukins (IL), interferon (IFN), tumor necrosis factor (TNF) [57], antimicrobial peptides (AMPs), T-cell markers and immunoglobulins are induced in a cascade as an immune response [1]. IRF3 is a key regulator of type I IFN gene expression in response to viral infection [1]. In the present study, *irf3* expression was also upregulated at 48 hpi, again indicating the host's attempt to control and eliminate the virus by inhibiting viral replication. The expression of *mx* and *ifn* genes in the brain of sea bream and seabass infected with NNV has been observed previously [14,58].

AMPs recruit effector cells to the site of infection [59,60]. They act against a wide range of pathogens, including non-enveloped viruses, such as NNV [61]. The C3 gene, which is an AMP, is part of the complement system and is directly involved in the host cell defense [62,63]. The expression of this gene increased significantly from 24 hpi, maintaining its expression until 144 hpi, suggesting that this gene is expressed early after the infection occurs and functions as an activator of the immune system [1]. Upregulation of *c3* in the gonad and brain of European seabass and Gilthead sea bream was observed in response to infection with NNV [59].

Adaptive immunity is dependent on the activation of T cells [64], which are activated by a costimulatory signal from antigen-presenting cells (APCs) [65]. This signal arises from the binding of CD80 and CD86 to CD28 [66,67]. *cd28* is expressed on the surface of most T cells, with higher values seen in activated cells [65]. Activation of CD28 promotes T-cell proliferation, which produces cytokines, such as interleukins, and aids cell survival [68]. The expression of this gene in the head kidney and brain of fish after infection with NNV has showed maximum expression at 24 hpi (first sampling time) and decreases thereafter [64]. We observed the expression of *cd28* to achieve its maximum at 9 hpi, reducing its values until 48 hpi; thereafter, *cd28* expression increased slightly until the end of the trial. Higher and lower *cd28* expression suggests that, as the NNV infection progresses, *cd28* expression tended to a stimulation of T cells and modulation of the metabolic processes that regulate the pro-inflammatory T-cell response [64].

NNV infection in European seabass triggers the expression of pro-inflammatory cytokines, such as TNFα and IL-1β. TNFα is an important mediator in resistance against bacterial and viral infections, regulating inflammation and apoptosis of target cells [56]. It is produced mainly by macrophages [69] and triggers the expression of numerous genes, such as IL-1β, a chemoattractant pro-inflammatory cytokine for leukocytes in fish [70,71]. It regulates immune responses and is an indicator of inflammation and neurodegeneration of the central nervous system [72]. These two pro-inflammatory cytokines can be rapidly secreted in response to a stimulus caused by an infectious agent [73]. In the present study, an increase in *tnfα* expression was recorded in the first few hours ( 6, 9 and 24) post-infection and at 144 hpi, coinciding with the increase in nitric oxide also at this time, thus corroborating their relationship. This is explained by the fact that this gene regulates inflammation at the beginning of the infection, as it increases the phagocytic activity of leukocytes in fish [70]. The expression of *il-1β* may have been triggered by *tnfα*, which was also seen to be expressed in the first few hours post-infection. However, this interleukin showed a high level of expression at 144 hpi relative to basal levels seen at 0 hpi, when high numbers of neutrophils and monocytes were present in the blood. The presence of such blood cells suggests the presence of high levels of viral particles, and these blood cells migrate to the site of infection, due to the stimulation of leukocytes by the infection [70].

Typical brain-tissue damage caused by NNV infection may be related to the overexpression of these cytokines [14] that is associated with the high level of mortality recorded in the present study.

## 5. Conclusions

Early detection of responses against *Betanodavirus* infections could help the industry deal with disease problems more effectively, and that was evident in European seabass juveniles in this trial. The early reduction of WBCs is a common response to infection and seemed to also occur during NNV infection, while the increased neutrophils at 144 hpi were accompanied by an increase in nitric oxide concentration produced by these cells. Furthermore, the antioxidant defenses started to decrease by 120 hpi, resulting in lipidic damage. Interaction between NNV and the immune responses could be seen within a few hours after infection, as verified by the expression of *c3* and *cd28*, and control of viral replication through the expression of *mxA* and *irf3*. The dynamics of the interaction between the virus and the host response could be seen over the time course of 144 hpi, coinciding with increased expression of *tnfα* and *il-1β*, which ultimately results in high levels of mortality. Acute infection was obtained with the virus dose used; however, a less severe infection in which fish can survive may help us to understand how the host's immune response can overcome the virus.

**Supplementary Materials:** The following supporting information can be downloaded at https://www.mdpi.com/article/10.3390/fishes7020063/s1. Figure S1: Standard curve of the mean Ct value of each of the samples in real-time reverse-transcriptase polymerase chain reaction (RT-qPCR), according to the 10 in 10 dilutions of known NNV TCID$_{50}$ concentrations.. Table S1: Genes related to the immune system of NNV-infected European seabass analyzed in real-time PCR [74–77].

**Author Contributions:** Conceptualization, A.P., J.Z.C., K.D.T. and T.B.; methodology, M.V. and T.B.; software, M.V. and R.P.; validation, A.P., J.Z.C., K.D.T. and T.B.; formal analysis, M.V., M.S. and R.P.; investigation, M.V., D.P., P.P., P.S., B.d.C., R.P. and T.B.; resources, A.P. and T.B.; data curation, M.V., D.P., P.P., P.S., B.d.C. and R.P.; writing—original draft preparation, M.V.; writing—review and editing, D.P., M.S., P.S., B.d.C., R.P., J.Z.C., K.D.T. and T.B.; visualization, M.V.; supervision, A.P. and T.B.; project administration, A.P. and T.B.; funding acquisition, A.P. and T.B. All authors have read and agreed to the published version of the manuscript.

**Funding:** This study was supported by Fundação para a Ciência e Tecnologia (FCT) through the strategic projects UIDB /04292/2020 and UIDP/04292/2020 granted to MARE and the project MAR-02.05.01-FEAMP-0013.

**Institutional Review Board Statement:** The study was conducted in accordance with "The guidelines on animals protection used for scientific purposes from the European Directive 2010/63/EU" under project permission 0421/000/000/2019.

**Informed Consent Statement:** Not applicable.

**Data Availability Statement:** Data regarding the results presented in this article will be made available upon reasonable request to the author.

**Conflicts of Interest:** The authors declare no conflict of interest. The funders had no role in the design of the study; in the collection, analyses or interpretation of data; in the writing of the manuscript; or in the decision to publish the results.

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
