# Peer review of "Early Immune Modulation in European Seabass (Dicentrarchus labrax) Juveniles in Response to Betanodavirus Infection"

_fishes, doi:10.3390/fishes7020063_

Round 1

Reviewer 1 Report

The paper describes the interaction between NNV infection and the several immune responses in the European seabass. The experiment data on early immune modulation via several factors are worthwhile. Even though it is known that it’s very difficult to find comprehensive linkages between immune responses in the aquatic organism (i.e., a viral infection of fish), I think some unnecessary sentences that ha nothing to do with your conclusion should be revised in the discussion section. Based on the “Conclusion” section, it would be better to revise the discussion section.

Other issues,

- L36: RGNNV is widely distributed worldwide including Southern European. I think more areas or regions should be added in the Introduction section.

- L59-60: It’s difficult to understand the meaning that you want to suggest in this part associated with sentences in L56-L58. I recommend rewriting. Or I think it is okay to delete this sentence in this part.

- L73: What water temp. and aqua-tank size (or vol.) for acclimation? And did you confirm the NNV-free before or during acclimation?

- L205: Italics – Dicentrarchus labrax

- L307: From the challenge test in this study, this suggestion might be progressive. Because the clinical signs of NNV infectious depend on several factors as you already described in the Introduction section. Thus, minimal infectious dose or other doses associated with mortality conditions should be discussed with references.  

- L417-419: I am not sure that meaning of this sentence. It is not insufficient in the discussion part in this article.

Author Response

L36 -  we included other areas

L59-60 - we delete the sentence as suggested by the reviewer

L73 - we included the water temperature and volume of the tank, and the collection of samples to confirm the NNV - free condition of the fish

L205 - corrected in the document

L307 - we change the sentence

L417-419 - revised the sentence.

Reviewer 2 Report

This manuscript describes the early response of European seabass (Dicentrarchus 2 labrax) juveniles to betanodavirus infection, by detecting the mortality, haematological parameters, viral loads, and gene expression. This paper is very descriptive based on speculation due to lack of substantial data.

  1. Lines 60-63, please give a detailed description of progress on NNV infection, in order to help readers understand the importance of this manuscript.
  2. Whereas statistical analysis was performed, whether these data could be replicated should be further confirmed. For example, it is very hard to understand why there was decreased expression of IRF3 during 9 to 24 post infection in Figure 4B. Same questions might be for Figure 2 and Figure 3, because the error bars were often so high.
  3. The authors stated IFN response is important for viral clearance. Why IFN genes were not detected in Figure 4?
  4. The section of Discussion was so long, which could be shortened by at least half.

Author Response

  1. L60-63 - the text was revised
  2. In the cases of the high values of error bars mainly observed in Figures 2 and 3, it's due to the variability found in the animals and also due to the number of animals sampled. The number of animals used is restricted due to ethical issues, reducing the number of fish but maintaining the statistical value.  However, when values were statistically different the differences were detected. IRF3 decreased from 6h until 24 (minimum).
  3. IFR3 is an interferon (IFN) type I molecule, that was detected and are included in Figure 4
  4. the discussion was shortened

Round 2

Reviewer 2 Report

  1. About the high values of error bars in Figure 2/3/4, the authors think that these are due to ethical issues. I think that the authors want to know whether these data could be replicated? It is very hard to understand why there was decreased expression of IRF3 during 9 to 24 post infection in Figure 4B. In response to viral replication, the IFN response must be initiated rapidly, so the host antiviral genes have to be rapidly activated or induced. However, why there is deceased expression of the IRF3 in the early stage of viral replication? These data are very contradictory to the understanding of fish IFN response, and I think it is of interest to further investigate. I suggest that the authors give a reasonable interpretation or provide further replicable data.
  2. In mammals, IRF3 is a constitutively expressed in virally infected cells. Unlike mammalian IRF3, fish IRF3 can be induced by viral infection or IFN, so it is a typical interferon stimulated gene (ISG). But it is not an INTERFERON gene. Detection of IFN gene expression is a direct evidence for the initiation of fish IFN response.
